# Stable Resistive Switching in ZnO/PVA:MoS_2_ Bilayer Memristor

**DOI:** 10.3390/nano12121977

**Published:** 2022-06-09

**Authors:** Tangyou Sun, Hui Shi, Shuai Gao, Zhiping Zhou, Zhiqiang Yu, Wenjing Guo, Haiou Li, Fabi Zhang, Zhimou Xu, Xiaowen Zhang

**Affiliations:** 1Guangxi Key Laboratory of Precision Navigation Technology and Application, Guilin University of Electronic Technology, Guilin 541004, China; suntangyou@guet.edu.cn (T.S.); shidada413@163.com (H.S.); gaoshuai202206@163.com (S.G.); wenjing@guet.edu.cn (W.G.); lihaiou@guet.edu.cn (H.L.); zhangfabi@outlook.com (F.Z.); 2State Key Laboratory of Advanced Optical Communication Systems and Networks, School of Electronics Engineering and Computer Science, Peking University, Beijing 100871, China; zjzhou@pku.edu.cn; 3School of Electrical and Information Engineering, Guangxi University of Science and Technology, Liuzhou 545006, China; 4School of Optical and Electronic Information, Huazhong University of Science and Technology, Wuhan 430074, China; xuzhimou@mail.hust.edu.cn

**Keywords:** resistive switching, ZnO/PVA:MoS_2_, data retention, endurance, memory window

## Abstract

Reliability of nonvolatile resistive switching devices is the key point for practical applications of next-generation nonvolatile memories. Nowadays, nanostructured organic/inorganic heterojunction composites have gained wide attention due to their application potential in terms of large scalability and low-cost fabrication technique. In this study, the interaction between polyvinyl alcohol (PVA) and two-dimensional material molybdenum disulfide (MoS_2_) with different mixing ratios was investigated. The result confirms that the optimal ratio of PVA:MoS_2_ is 4:1, which presents an excellent resistive switching behavior. Moreover, we propose a resistive switching model of Ag/ZnO/PVA:MoS_2_/ITO bilayer structure, which inserts the ZnO as the protective layer between the electrode and the composite film. Compared with the device without ZnO layer structure, the resistive switching performance of Ag/ZnO/PVA:MoS_2_/ITO was improved greatly. Furthermore, a large resistive memory window up to 10^4^ was observed in the Ag/ZnO/PVA:MoS_2_/ITO device, which enhanced at least three orders of magnitude more than the Ag/PVA:MoS_2_/ITO device. The proposed nanostructured Ag/ZnO/PVA:MoS_2_/ITO device has shown great application potential for the nonvolatile multilevel data storage memory.

## 1. Introduction

Recently, random resistive access memory (RRAM) with outstanding characteristics of a fast operation speed, high storage density, low power consumption, long retention time, multilevel data storage and simple structure has become the research hotspot for the next-generation nonvolatile memory applications [1,2,3]. RRAM has a typical sandwich structure with metal–insulator–metal, where various materials have been used as the switching layer. According to the basic properties of the dielectric material layer, it can be divided into inorganic materials and organic materials [4]. Owing to the characteristics of organic materials, including easy modification, large scalability, simple preparation method and so on, there is a growing demand for organic-based resistive switching memory, such as poly(4-vinylphenol) (PVP) [5] and polyvinyl alcohol (PVA) [6,7]. Metal oxides consisting of Al_2_O_3_ [8], TiO_2_ [9] and ZnO [10,11] have been widely studied as inorganic materials due to their simple structure, easy control of material composition and compatibility with CMOS technology. Two-dimensional (2D) materials such as graphene [12], MoS_2_ [13,14], WS_2_ [15] and h-BN [16] have gained great attention owing to their excellent electrical and mechanical characteristics. At present, the discussion of RRAM mainly focuses on addressing the major key challenges, which are retention, endurance, ON/OFF ratio and operating voltage. Scalability, variability and low-cost fabrication are other challenges of concern for RRAM. However, the resistive switching (RS) performance of mono-components is not ideal because of their own physical limitations. For example, organic polymers often struggle to obtain stable RS memory on account of its environmental sensitivity [17]. Table 1 summarizes the research results of RRAM over the years. It can be seen that the performance of mixed layer and multi-layer structures is better than that of a single traditional device.

What is more, some studies have reported that by effective mixing of the 2D inorganic materials and organic matters to produce a nanostructured organic/inorganic composite dielectric layer, a decent property of the RRAM devices can be obtained. For instance, Lv et al. [24] studied the resistive memory with the resistive layer related to polymer [6,6]-phenyl-C61-butyric acid methyl ester (PCBM)−MoS_2_, which presents both WORM and flash memory characteristic with superior electrical biostability. In addition, the stacking of the structure by inserting a metal oxide between the inorganic/organic composites and electrode can effectively enhance the RS performance of the RRAM devices. Varun et al. [5] obtained a resistive memory with low power consumption and high storage window by combining PVP: GO composite film with ultra-thin HfO_x_. Liao et al. [25] combined PDPPBTT with ZnO, proposed that the resistive switching mechanism of the device is related to the formation and rupture of conductive filaments formed by conductive ions and further demonstrated the application potential of Ag/PDPPBTT/ZnO/ITO devices in the field of biological synaptic simulation in the future.

In this work, we fabricated a memory device based on the ZnO and PVA:MoS_2_ composite film stacked structure in which PVA:MoS_2_ acts as organic heterojunction and inorganic ZnO acts as function layer. The RS performance of PVA and MoS_2_ under different mixing ratios has been studied systematically. By inserting the ZnO layer between electrode and composite under the optimal mixing ratio, the non-volatile memory characteristics of the device were observed, including a larger resistive memory window, better repeatability and a longer retention performance. What is more, a stable middle state of the device can be obtained after inserting ZnO as a voltage divider, which demonstrates that the Ag/ZnO/PVA:MoS_2_/ITO device has important application potential for next-generation nonvolatile high-density data storage memory.

## 2. Experiment

MoS_2_ suspension (1 mg/mL) was purchased from XF Nano Co. Ltd. (Nanjing, China). PVA powders were procured from Sigma-Aldrich (Shanghai, China). Indium tin oxide (ITO) with 150 nm-thick coated glass substrates were obtained from South China Science & Technology Company Limited (Hunan, China). To achieve a PVA:MoS_2_ nanocomposite film, a transparent PVA solution was prepared by adding 0.6 g PVA powder into 10 mL deionized water (DI). The mixture was magnetically stirred for 2 h at the temperature of 60 °C and then allowed to cool down to room temperature. Subsequently, the mixtures of PVA solution and MoS_2_ suspension with the volume ratio of 4:1 and 4:3 were obtained after vigorous stirring for 12 h at room temperature, respectively. The graphical ITO bottom electrode (BE) with a line width of 100 μm was produced by photolithography and wet etching process. The as-prepared mixture of PVA:MoS_2_ was spin-coated on the graphical ITO bottom electrode, which was then dried on the hot plate at 80 °C for more than 2 h to form a PVA:MoS_2_ composite film with a thickness of ~150 nm. Next, a 10 nm thick ZnO layer was deposited on the PVA:MoS_2_ composite film surface by magnetron sputtering. After that, a Ag top electrode (TE) with a thickness of 150 nm was sputtered via a metal shadow mask with a typical device size of 10 μm × 10 μm. For the device performance comparison, the Ag/PVA:MoS_2_/ITO device and Ag/PVA/ITO device were also fabricated, respectively. The electrical characterization of as-prepared devices was performed using a Keithley2636 Source Meter with biasing at TE, while the BE was kept at ground potential.

## 3. Results and Discussion

Figure 1a shows the schematic illustration of Ag/ZnO/PVA:MoS_2_/ITO resistive memory, which is a cross-point structure. The scanning electron microscope (SEM) surface diagrams of the ZnO layer and PVA:MoS_2_ layer show that the ZnO and both the PVA:MoS_2_ are well covered on the bottom electrode ITO. X-ray photoelectron spectroscopy (XPS) analysis of the ZnO layer was performed as shown in Figure 1b. The Zn2p core-level spectra can be fitted by two distinct peaks such as the Zn2p_1/2_ and Zn2p_3/2_, corresponding to the 1021.55 eV and 1044.56 eV, respectively, which reveals the strong bonding between Zn atoms and oxygen ions and a perfect stoichiometry of the ZnO film [26]. Figure 1c shows the XPS spectra of the O1s, which can be deconvoluted into two peaks related to the lattice oxygens and non-lattice oxygens [11]. It can be seen that binding energy peaks at 530.23 eV are attributed to the oxygen ions, while higher energy peaks at 531.76 eV are corresponding to oxygen vacancies in the ZnO layer [27]. Figure 1d shows the measured Raman spectra of MoS_2_. It is clear that two obvious characteristic peaks at 381.09 cm^−1^ and 407.56 cm^−1^ can be found, which are MoS_2_ lattice vibration modes caused by covalent bond stretching of Mo and S, corresponding to plane Mo and S atomic vibration modes ($E_{2g}^{1}$) and S atomic vibration modes ($A_{1g}^{1}$), respectively. Additionally, it proves the material is multi-layer stacked MoS_2_ [28,29]. In order to confirm the phase composition and microstructure of PVA:MoS_2_ composites, the phase analysis of the PVA:MoS_2_ film was carried out by X-ray diffraction (XRD) (Figure 1e). It is obvious from Figure 1e that the diffraction peak of pure PVA is about 2θ = 23° [30]. The diffraction peak of PVA hardly shifts but the intensity of the peak decreases with the increase in MoS_2_, indicating that the crystallinity of the PVA polymer decreases [31].

The current–voltage (*I*–*V*) characteristic was studied to examine the resistive switching behavior of the fabricated devices as shown in Figure 2, and the electrical tests of all devices are carried out at a normal temperature and pressure. The sweeping voltages optimizing the uniformity of the device were applied on the top electrode with the bottom electrode grounded and the voltage loop was 0 V→*V*_max_→0 V→−*V*_max_→0 V. A compliance current of 0.1 mA was employed to prevent breakdown of the devices. The resistive memory behavior can be described by the SET and RESET switching voltages. While the positive voltage sweeps from 0 to *V*_max_ and reaches a voltage named *V*_set_, the device switches from an initial high-resistance state (HRS) to a low-resistance state (LRS), which was labeled as the “SET” process. The device maintains in the LRS before the reverse voltage is applied and the device transitions from LRS to HRS when the voltage reaches *V*_reset_, corresponding to “RESET” process. Figure 2a shows the *I*–*V* curve of the Ag/PVA:MoS_2_/ITO (PVA:MoS_2_ = 4:1) device, which exhibits an obvious hysteresis under forward-bias voltage and reverse-bias voltage, representing a typical bipolar resistive characteristic. The voltage increased gradually from 0 to 3 V and then a sharp rise in current was observed at *V*_set_ = 0.91 V, implying the formation of conductive filaments (CFs) between two electrodes. It is interesting to note that there was no electronic forming process here, which was beneficial for scaling down the complexity of the memory circuit [32,33]. After that, the negative bias was back from −3 V to 0 and a dramatic drop in current can be surveyed when the voltage reached *V*_reset_ = −0.79 V, suggesting the collapse of conductive filaments between two electrodes.

The insect picture in Figure 2a shows the electrical characteristics of the Ag/PVA/ITO device (PVA:MoS_2_ = 4:0). It is obvious that no bipolar resistive switching characteristic can be observed, indicating that the RS behavior in hybrid PVA:MoS_2_ is related to the local high-electric field generated at the edge of the MoS_2_ nanosheet consisting of sulfur vacancy. Figure 2b shows the *I*–*V* characteristics of the Ag/PVA:MoS_2_/ITO device with PVA:MoS_2_ = 4:3. By comparing the *I*–*V* curves of the as-prepared devices with different concentrations, it is demonstrated that the decreasing crystallinity puts on the proportion of amorphous part with the increase in MoS_2_ content, which is beneficial for the migration of conductive ions. Hence, the semiconductor properties of MoS_2_ dominates the electrical properties of the nanocomposite when the concentration of MoS_2_ increases and the leakage current increases, resulting in the deterioration of the memory window [34]. Therefore, the optimal volume ratio of the composite is PVA:MoS_2_ = 4:1, which displays an excellent resistive switching behavior in this work and would be considered for subsequent studies.

MoS_2_ nanosheets mixed with PVA can improve the memory performance; however, the device may still have the problem of poor stability. Figure 2c,d show the cycling endurance and retention time characteristics of Ag/PVA:MoS_2_/ITO (/PVA:MoS_2_ = 4:1), respectively, where the read voltage is 0.2 V. It can be seen that the memory window of the Ag/PVA:MoS_2_/ITO devices is about 10 and the device failure occurs after about 380 cycles and 800 s, which indicates that the stability of organic memory cannot be reliably maintained. The main issue in the hybrid device is the highly localized electric field around MoS_2_ nanosheets [7], which might have induced the failure of device performance.

In order to solve this problem, ZnO is selected as the protective layer to sustain the majority voltage, which would reduce the local electric field. Figure 3a reveals the *I*–*V* curve of the Ag/ZnO/PVA:MoS_2_/ITO device under different cycle times, which exhibits a reliable nonvolatile switching behavior. By comparing the *I*–*V* curves of Ag/ZnO/PVA:MoS_2_/ITO, Ag/PVA:MoS_2_/ITO and Ag/PVA/ITO devices, it is found that the RS performance of the Ag/ZnO/PVA:MoS_2_/ITO device is significantly enhanced due to the development of the highly localized electric field of nanocomposite. Moreover, there is a stable intermediate resistance state as displayed in Figure 3a, which demonstrates that the Ag/ZnO/PVA:MoS_2_/ITO device has the application potential of high-density multilevel data storage.

The cumulative probability distribution of *V*_set_ and *V*_reset_ voltages of the Ag/PVA:MoS_2_/ITO and Ag/ZnO/PVA:MoS_2_/ITO devices has been shown in Figure 3b. Although the inserted ZnO layer increases the SET and RESET voltage of the device, the uniformity of the switching voltage can be significantly improved, which indicates that the ZnO has been served, as the divider stands most of the electric field, reduces the instability of the nonvolatile resistive switching stemming from the large localized electric field and optimizes the uniformity of the device. The endurance test of Ag/ZnO/PVA:MoS_2_/ITO can be depicted as in Figure 3c, and it presents the reliable performance over 1000 switching cycles. Meanwhile, the water and oxygen chemisorbed by the organic active layer will also lead to the degradation of the RS performance and stability of the device [35]. As shown in Figure 3d, after the ZnO was inserted as a protective layer, the LRS and HRS of Ag/ZnO/PVA:MoS_2_/ITO were maintained for more than 5000 s without any obvious deterioration. The resistive memory window of the device can increase to ~10^4^, which is improved by at least three orders of magnitude compared with the Ag/PVA:MoS_2_/ITO.

In order to explore the conduction mechanism, the piecewise linear fitting of the *I*–*V* curve plotted in log-log scale was carried out to analyze the resistive switching behavior of the Ag/ZnO/PVA:MoS_2_/ITO device as depicted in Figure 4a. During the SET process, the conduction mechanism follows Ohm’s law with the fitting slope of about 1.06, and the *I*–*V* curve shows a linear relation (I∝V) in a low voltage area (Region A), suggesting that the device has been governed by thermal excitation produced by the trap-filled limited minority carrier in the HRS. The slope of the *I*–*V* curve increases to ~2.37 (Region B), which agrees with the nonlinear behavior with the Child conduction mechanism ($I\propto V^{2}$). At this stage, the current is completely controlled by the space charge, which restricts the further injection of carriers into the dielectric layer. When the voltage increases to *V*_set_, all traps are filled and a large number of free carriers increase dramatically, resulting in a sudden surge of current, and the device turns from HRS to LRS. Thus, the electrical behaviors of the device are ascribed to the conduction mechanism of the trap-filled space charge limited conduction (SCLC) [36] in the HRS. The relation between the current density and the bias voltage can be defined as follows:$$J=9\varepsilon_{m}\mu \theta V^{2}/8d^{3}$$
where J is the current density, $\varepsilon_{m}$ is the dielectric constant, μ is the mobility of charge carrier, θ is the ratio of free electrons to trapped electrons, *V* is the bias voltage and d is the distance between the two electrodes [37].

For the LRS, the slope of the curve is about 1.07 (Region C) and the conductive phenomenon conforms with the Ohmic conduction mechanism, demonstrating that the metallic conductive filaments are formed inside the device. The different conduction properties of Ag/ZnO/PVA:MoS_2_/ITO in high- and low-resistance states indicate that the resistive switching behavior of the device follows the conduction mechanism adjusted by the metallic conductive filaments [38].

The switching behavior can also be explained through the energy band diagram of the Ag/ZnO/PVA:MoS_2_/ITO device, as presented in Figure 4b. When the voltage is applied to the Ag electrode, electrons can easily pass through the ZnO due to the work functions of Ag (4.2 eV) and ZnO (5.2 eV) being approximate [39]. The highest occupied molecular orbital (HOMO) and lowest unoccupied molecular orbital (LUMO) of PVA and MoS_2_ are shown in Figure 4b [40,41,42,43]. A large energy gap between MoS_2_ and PVA suggests that MoS_2_ acts as a trapping centra and the electron transfer will occur from PVA to MoS_2_. It also indicates that the conductive mechanism in the HRS is SCLC. When the lower reverse voltage is applied, the conductive filament does not break, which confirms the device is nonvolatile. A number of sulfur vacancies acting as the trap centers are introduced by the MoS_2_ nanosheets, which contribute to the formation of conductive filaments when the flow of current is highly localized to a small fraction of device area, resulting in the RS behavior [44]. Thus, the memory characteristic in PVA:MoS_2_ composite is related to the highly localized electric field formed by MoS_2_ nanosheets, and the local electric field inside the PVA:MoS_2_ film can be reduced by inserting the ZnO protective layer as shown in Figure 4(ci), improving the stability of the device.

According to the above discussion, which demonstrates the device is a CFs-type memory, a nonvolatile resistive switching model has been proposed to analyze the RS phenomena of the Ag/ZnO/PVA:MoS_2_/ITO device as shown in Figure 4b. The initial state of the device is HRS. Then, the conduction behavior of the Ag/ZnO/PVA:MoS_2_/ITO device follows Ohmic conduction mechanism in the LRS, indicating the formation of a metallic conductive path in the Ag/ZnO/PVA:MoS_2_/ITO device. When the positive voltage is applied to the top Ag electrode, the Ag loses electrons and would be oxidized into Ag^+^ ions. Then, the Ag^+^ ions would migrate toward the ITO electrode under the high external electric field. Finally, the Ag^+^ ions would be reduced to Ag atoms near the ITO electrode. Once the Ag conductive filament connects the two electrodes with the accumulation of Ag atoms from the top to bottom electrodes, the device would change from HRS to LRS. The existence of ZnO can also lead to the generation of some oxygen vacancies, which is also beneficial for the formation of conductive filaments. However, the conductive filaments in our device are more likely to be Ag filaments [10,25,45,46]. When the voltage polarity is reversed, the Ag conductive filament will break at the weakest point and the device will return to the high resistance state (HRS). Figure 4(cii,ciii) show the schematic presentation of resistive switching mechanism in LRS and HRS, respectively.

## 4. Conclusions

In summary, the Ag/ZnO/PVA:MoS_2_/ITO organic/inorganic heterojunction resistive memory has been designed by a spin-coating method and other simple processes in this paper. The effects of the PVA/MoS_2_ mixtures with various volume ratios on the RS performance of devices were studied systematically. The RS characteristics of Ag/ZnO/PVA:MoS_2_/ITO devices were discussed and the physical model of the switching mechanism was established. The XRD analysis shows that the addition of MoS_2_ will reduce the crystalline state of the switching layer and produce the RS performance. It also concludes that the Ag/PVA:MoS_2_/ITO device with the optimal ratio PVA:MoS_2_ = 4:1 shows excellent resistive switching behaviors, including a high resistive memory window up to 10^4^, reliable repeatability and long cycling time. This work suggests that the Ag/ZnO/PVA:MoS_2_/ITO device has the application potential of high-density multilevel data storage. 

## Figures and Tables

**Figure 1 nanomaterials-12-01977-f001:**
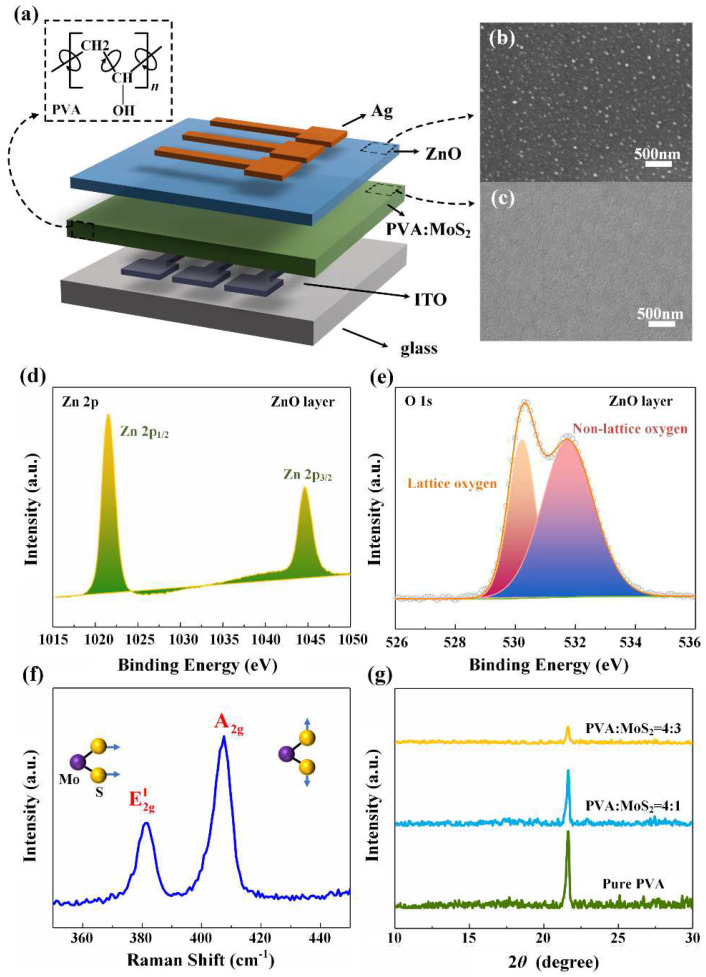
Structure and basic characteristics of the device. (**a**) Schematic illustration of the Ag/ZnO/PVA:MoS_2_/ITO memory, (**b**) SEM image of the ZnO surface, (**c**) PVA:MoS_2_ deposited on the ITO glass substrate, (**d**) XPS spectra of Zn2p in the ZnO layer, (**e**) XPS spectra of O1s in the ZnO layer, (**f**) Raman spectra of MoS_2_, (**g**) XRD patterns of PVA:MoS_2_ composite.

**Figure 2 nanomaterials-12-01977-f002:**
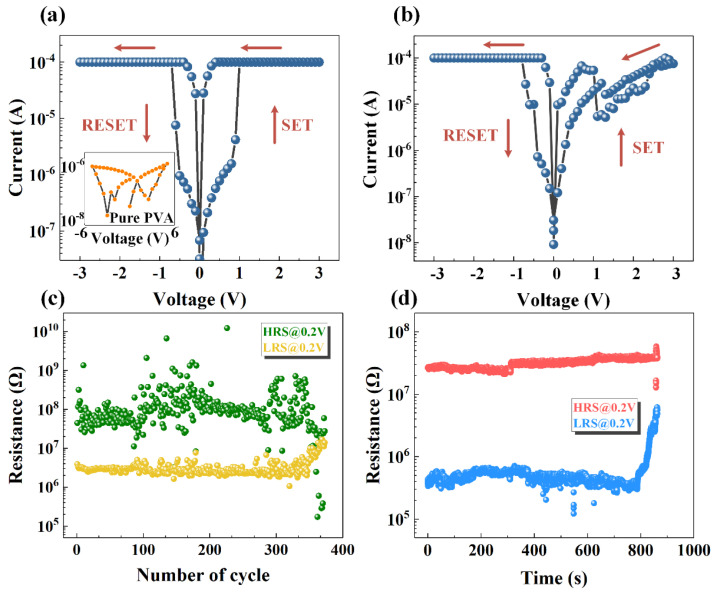
The *I*–*V* curve and switching behaviors of (**a**) Ag/PVA:MoS_2_/ITO, PVA:MoS_2_ = 4:1, insect: Ag/PVA/ITO, (**b**) Ag/PVA:MoS_2_/ITO, PVA:MoS_2_ = 4:3. (**c**) The endurance of PVA:MoS_2_ = 4:1 RRAM, (**d**) The resistive-state retention time of HRS and LRS of Ag/PVA:MoS_2_/ITO, PVA:MoS_2_ = 4:1.

**Figure 3 nanomaterials-12-01977-f003:**
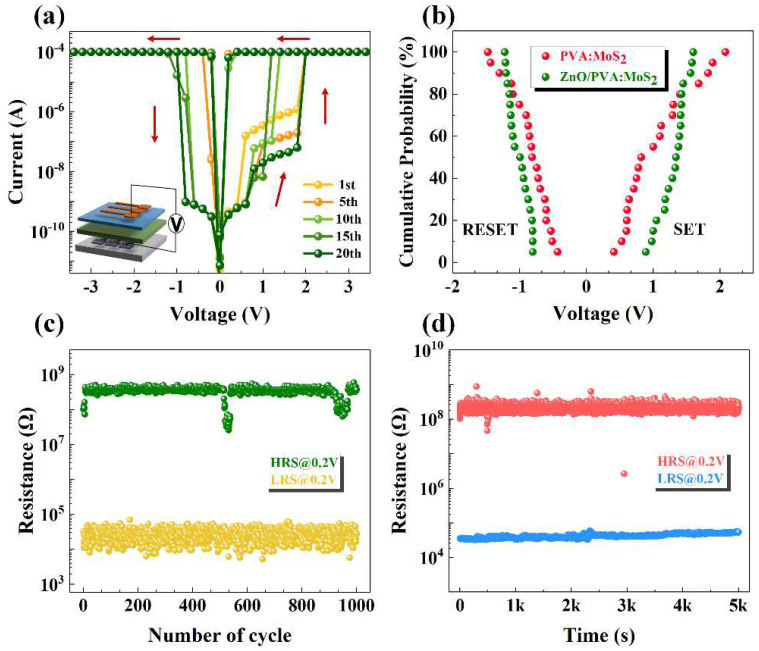
(**a**) The *I*–*V* characteristics of the Ag/ZnO/PVA:MoS_2_/ITO, PVA:MoS_2_ = 4:1, insert: electrical circuit for the device. (**b**) Cumulative probability of SET/RESET voltages for Ag/PVA:MoS_2_/ITO and Ag/ZnO/PVA:MoS_2_/ITO, PVA:MoS_2_ = 4:1. (**c**) DC sweep mode endurance cycles at read voltage 0.2 V, respectively. (**d**) Retention property in both LRS and HRS at read voltage 0.2 V.

**Figure 4 nanomaterials-12-01977-f004:**
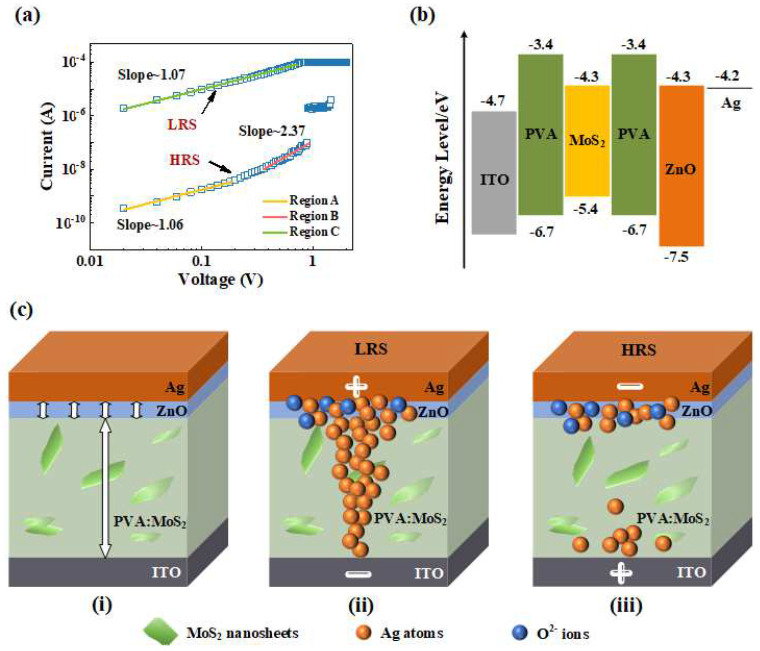
(**a**) *I*–*V* curve of the Ag/ZnO/PVA:MoS_2_/ITO memristor plotted in log-log scale, (**b**) Energy band diagram of Ag/ZnO/PVA:MoS_2_/ITO device structure, (**c**) Schematic presentation of resistive switching mechanism in Pristine (**i**), LRS (**ii**) and HRS (**iii**), respectively.

**Table 1 nanomaterials-12-01977-t001:** Summary of some recent research on RRAM devices.

Structure	Deposition Technique	ON/OFF Ratio	Endurance Cycles	Retention Time(s)	Year	Ref.
Ag/TiO_2_/FTO	Hydrothermal method	~10×	16	-	2019	[18]
Ag/MoS_2_/PMMA/SiO_2_	Sputtering	10^4^	300	10^3^	2020	[19]
Ag/ZnO/FTO	CVD	>50	~40	>10^3^	2020	[10]
Au/TiO_2_/ZrO_2_/ITO	Spin coating	10^2^	100	-	2020	[20]
Ti/h-BN/Au	CVD	-	120	10^3^	2021	[21]
Ag/PVA:ZnO/FTO	Spin coating	~10^3^	-	-	2021	[22]
Ag/GO:PVP/TiO_x_/ITO	ALD, Spin coating	>10^3^	128	>7 × 10^3^	2022	[23]
Ag/ZnO/PVA:MoS_2_/ITO	Sputtering, Spin coating	~10^4^	>1000	>5 × 10^3^	-	This work

## Data Availability

Data is contained within the article.

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
