# Peer review of "Stable Resistive Switching in ZnO/PVA:MoS2 Bilayer Memristor"

_nanomaterials, 2022, doi:10.3390/nano12121977_

Round 1

Reviewer 1 Report

The authors  claims organic/inorganic heterojunction composites (Ag/ZnO/PVA:MoS2/ITO) to show excellent memristive properties. However, it is hardly seen supportive data for reliable switching.

  1. Figure (a) and (c) do not show the same current level.
  2. The authors claimed large on/off ratio with additional ZnO layer in figure 3(d) due to voltage divide effect. However, usually with additional layer (or resistor), the on/off ratio is decreased due to low voltage on the active switching layer. 
  3. In line 172 and title, the authors claimed "highly reliable" switching. However, in figure 2(c),  and figure 3, the device does not show any improved performance compared to conventional inorganic memristors. The data was analyzed quantitatively, the authors should compare their result with other conventional memristors qualitatively. I agree that 4:1 device shows reliable switching compared to other composition, but the device performance with 4:1 is also not reliable. Also, the authors should provide why highly reliable performance can be possible with the device in terms of structure or materials. I do not see any points for performance improvement.
  4. Any evidence on local electric field in line 175?
  5. in line 184, the authors mentions oxygen leading to the degradation of the film and the ZnO can be used as a protective layer. However, ZnO also has oxygen components. The authors should provide the mechanism why ZnO can be used as a protective layer.
  6. The authors should provide why conduction mechanism is analyzed in the manuscript. I cannot find any connection between conduction mechanism and device performance improvement from the manuscript. 

Reviewer 2 Report

Dear Editor;

The authors claimed that ZnO layer helps to establish electrical switching with memristive action in Ag/ZnO/PVA/MOS2/ITO composites, however, given the few data they have presented (especially in Fg2) it is not clear that this is the case here. The interpretation of electrical switching (set and reset) is not convincing and must be improved. The band structure in memristive matrix is highly dynamics due to ions transport and can’t be taken in a simple way as depicted in Fig. 4(a) to explain transport. The variance and asymmetry around zero bias in set and rest must also be addressed.

Here I highlight some of the important points for authors attention:

  1. How the thickness of the device effects the overall stability? Did the authors control the thickness so that the thickness of the memristive matrix is the same for all devices?
  2. The Ag electrodes isn’t protected from the environment, which can cause device deteriorations as well.
  3. Device with MOS2 must be checked separately (Ag/MoS2/ITO) to see any possible switching in this composite.
  4. How the authors limit the value of electrical current? It seems the value of 100 microamp is due saturation of the preamplifier? Could that destroy the device after just few 10 of cycles?
  5. Please draw the electrical circuit for the device.
  6. How the devices were initialized?

I think the authors need to address the above issues before the paper can be consider for publication in Nanomaterials.

Reviewer 3 Report

Comments

As the author claims that by using ZnO layer as heterostructure the performance of device improved a lot but following points are not address in the manuscript:

  1. In XRD graph there is no peak of ZnO and MoS2 are observed, and strong analysis is required.
  2. As the author claims best heterostructure device but it lacks Raman study.
  3. The XPX analysis is not properly discussed like before and after heterostructure effect of oxygen vacancies.
  4. In the I-V graphs (Figure2a,c) compliance current are seen in both Set and Reset. Why?
  5. The cycles of nanocomposite device are not compared with the heterostructure device.
  6. The I-V graphs do not support the endurance and retention graphs.
  7. In the keywords, author wrote “stacked structure” which can be replaced by material name of the active layer in the paper.
  8. On line no. 134, it’s mentioned that voltage increased gradually from 0 to 3V, but according to IV curves, the gradual increase in the voltage from 0 to 0.8V and then it is abrupt switching.
  9. The manuscript required English revision.
  10. For appropriate significance readership it is better to compare the memory results with other 2D materials in introduction. See the articles as follow:

A. (Neuro-Transistor Based on UV-Treated Charge Trapping in MoTe2 for Artificial Synaptic Features. 2020)

 B. (MoS2/Polymer Heterostructures Enabling Stable Resistive Switching and Multistate Randomness. 2020)

Round 2

Reviewer 1 Report

To say "highly reliable resistive switching", the uniformity is the most important factor. The main problem of memristor to be commercialized is uniformity. To claim highly reliable memristor, the authors should compare the uniformity with other devices, not with the control sample. I suggest removing "highly reliable" from the title so that the readers are not confused about the paper content.

Author Response

Thank you for your very professional and in-depth comments. We have changed our title to "Stable Resistive Switching in ZnO/PVA: MoS2 Bilayer Memristor".

Reviewer 3 Report

The author's response is satisfactory. I would recommend this manuscript for publication. 

Author Response

Thank you.